# Application of Emulsion Gels as Fat Substitutes in Meat Products

**DOI:** 10.3390/foods11131950

**Published:** 2022-06-30

**Authors:** Yuqing Ren, Lu Huang, Yinxiao Zhang, He Li, Di Zhao, Jinnuo Cao, Xinqi Liu

**Affiliations:** 1National Soybean Processing Industry Technology Innovation Center, School of Food and Health, Beijing Technology and Business University (BTBU), Beijing 100048, China; ryq512@163.com (Y.R.); huanglulu1119@163.com (L.H.); zhangyx268825@126.com (Y.Z.); zhaodi22121@163.com (D.Z.); 2Plant Meat (Hangzhou) Health Technology Limited Company, Hangzhou 310000, China; jinnuocao@163.com

**Keywords:** emulsion gel, fat substitute, double emulsion, gelled double emulsion, healthier meat products, reduced fat

## Abstract

Although traditional meat products are highly popular with consumers, the high levels of unsaturated fatty acids and cholesterol present significant health concerns. However, simply using plant oil rich in unsaturated fatty acids to replace animal fat in meat products causes a decline in product quality, such as lower levels of juiciness and hardness. Therefore, it is necessary to develop a fat substitute that can ensure the sensory quality of the product while reducing its fat content. Consequently, using emulsion gels to produce structured oils or introducing functional ingredients has attracted substantial attention for replacing the fat in meat products. This paper delineated emulsion gels into protein, polysaccharide, and protein–polysaccharide compound according to the matrix. The preparation methods and the application of the three emulsion gels as fat substitutes in meat products were reviewed. Since it displayed a unique separation structure, the double emulsion was highly suitable for encapsulating bioactive substances, such as functional oils, flavor components, and functional factors, while it also exhibited significant potential for developing low-fat or functional healthy meat products. This paper summarized the studies involving the utilization of double emulsion and gelled double emulsion as fat replacement agents to provide a theoretical basis for related research and new insight into the development of low-fat meat products.

## 1. Introduction

Fat-rich meat products are popular with consumers due to their sensory properties. Fat is closely related to several food characteristics, such as texture, taste, and appearance. It also acts as a structuring and tasting agent, supplies energy, serves as a carrier [1,2,3], and directly affects the quality of food consumption and consumer satisfaction. Although traditional meat products are important sources of high-value animal protein [4], the majority of the animal fat is rich in saturated fatty acids (SFA) and cholesterol, and excessive intake increases the incidence of cardiovascular disease, raising the concern for human health [5].

Frankfurters, bologna sausage, beef patties, and other popular meat products typically contain 20–30% fat [6]. The WHO recommends an SFA level of 10% of the total fat intake and that the dietary fat consumption should account for 15% to 30% of the total dietary energy [7]. Although most consumers have the perception of reducing fat and cholesterol intake, which reduces the sales of high-fat foods, it is not feasible to sacrifice the product quality to reduce the fat content of the food [8]. Furthermore, considering global environmental challenges, public health problems, sustainable development, and animal welfare issue [9], while ensuring that the food quality is not lower than the acceptable range for consumers, reducing the amount of animal fat in food and developing fat-free and low-fat food has become an urgent problem to be solved. Fat reduction can be achieved by incorporating fat substitutes in meat products.

Adding fat substitutes to meat products to improve the fatty acid proportions can benefit consumer health. Adding fat substitutes can reduce the fat content in meat products and enhance the distribution of fatty acids. Since the fatty acid composition and proportion significantly impact human health, the polyunsaturated fatty acid (PUFA) to SFA ratio should be between 0.4 and 1.0. Furthermore, because unsaturated fatty acids also have substantial health implications, it is recommended that the n-6/n-3 PUFA ratio not exceed 4. A high n-6/n-3 PUFA ratio can promote the incidence of cardiovascular disease, inflammation, and other disorders. Some meat, such as pork, presents suboptimal fatty acid ratios [7,10].

Using plant oil rich in unsaturated fatty acids to replace animal fat abundant in SFAs is currently attracting significant research attention. However, simply using plant oils to replace animal fats causes a decline in product quality, since animal fats display a solid, elastic structure at room temperature absent in liquid oils. The adipose tissue in meat products consists of liquid oil and solid fat in the connective tissue network, exhibiting both plastic and elastic properties. Animal fat particles play a critical role in the cooking loss rate, hardness, texture, juiciness, flavor, and appearance of meat products [11]. Therefore, liquid plant oil must be treated to resemble animal adipose tissue, reducing the fat content while preserving the original sensory properties of the product as much as possible [12].

Recent studies have focused on utilizing emulsion gels to substitute the animal fat in meat products, forming a network structure via carbohydrate or protein cross-linking, which acted as a matrix for structural fat simulation. Emulsion gel formation is considered a strategy for oil stabilization and structuring, presenting advantages such as transporting functional components and improving the sensory and physical product properties. Since emulsion gels are highly suitable for developing healthy meat products, many studies are currently examining their application prospects. Double emulsion displays potential for active substance encapsulation and can also be used to formulate fat substitutes. Although some studies have investigated the application of double emulsions and the utilization of double emulsion gel as a fat substitute in meat products, the research is still immature.

## 2. Emulsion Gel

### 2.1. A Basic Introduction of Emulsion Gel

An emulsion is a colloidal dispersion generated by a liquid in the form of small droplets distributed in another insoluble liquid [13]. Emulsion gels, soft solid materials, are emulsion with a gel network structure and stable mechanical properties [14], while the emulsified droplets are embedded in the gel matrix, making it a complex colloidal material that can exist in both emulsion and gel states [15]. The properties of emulsion gels result from complex interactions between their components [16]. In oil-in-water (O/W) emulsion gels, this colloidal structure can be formed either via the dispersion of the emulsion droplets in a continuous gel matrix or by the aggregation of the dispersed droplets in the particle gels [17]. Compared with standard emulsions, emulsion gels display better storage stability and the potential to prolong intestinal drug release [18]. They also exhibit excellent stability, which is often used for embedding flavor substances. Furthermore, emulsion gels present a soft solid texture and can exhibit physical behavior similar to fat in emulsified meat products [17], which is especially suitable for designing and developing healthy, functional foods while showing significant potential for utilization as fat substitutes to produce low-fat meat products. These gels can replace animal fat while maintaining the physical and chemical properties (especially stiffness and water-holding properties) of the products [18]. The use of plant oil rich in PUFAs for preparing solid-structured oils and fats for replacing animal fats has attracted increasing attention. Emulsion gels are widely used in meat products [19].

Emulsion gels typically use a stable protein emulsion as a matrix, while the excellent emulsification characteristics of proteins allow them to be adsorbed on the oil–water interface to stabilize the oil droplets [20]. In emulsion-droplet-filled gels, the continuous phase of the stable emulsion forms a gel network structure in certain induced conditions, while the dispersed phase fills the gel network to yield a soft solid material and create an emulsion gel. (Figure 1).

### 2.2. Preparation of the Emulsion Gel

The emulsion gel preparation process mainly consists of two steps: stable emulsion preparation and gel formation. After adding an emulsifier, water and oil are mixed via high-pressure homogenization or high-speed mixing [21] to form stable emulsions. The oil to water ratio and protein content are critical. Sufficient protein coverage stabilizes the oil–water interface, while too little protein may lead to the adsorption of multiple droplets on one protein molecule (bridging flocculation), facilitating emulsion creaming. However, an excess may cause the protein covering the droplet surface to be ejected from the surface junction zone (depletion flocculation), promoting droplet coalescence [22].

Obtaining a solid-like emulsion gel involves continuous phase gelation (emulsion-droplet-filled gels) or droplet aggregation (emulsion-droplet-aggregated gels). The second step involves inducing the stable emulsion to an emulsion gel [21], which is achieved in two ways. One requires heat induction using traditional heat treatment methods, while the other uses gelation agents to obtain a cold-set gel. Cold-induced gelation can be broadly divided into acid-, salt-, and enzyme-induced methods depending on the added gelation agents. Furthermore, regulating environmental and processing factors (including temperatures, pH, or ionic strength) is also essential for preparing emulsion gels [23].

Heat treatment (>65 °C) is a simple, rapid method commonly used for converting protein-stabilized emulsions into gels. During the heat process, the folded structures of the protein molecules open due to denaturation and aggregation, allowing the unfolded protein to form new intra- and interchain disulfide bonds. In appropriate conditions, the gel networks aggregate to form three-dimensional (3D) structures via the interaction between chemical forces, such as disulfide bonds, hydrogen bonds, and the hydrophobic interaction between molecules [24]. Extending the heating time deforms the protein in the gel network, increasing the gel strength and elasticity. However, the degree of denaturation and aggregation can also affect the physical properties of the final gel. Therefore, many factors must be considered when preparing gels via heat treatment, such as protein concentration, temperature, and heating time [25].

Cold-set gel preparation involves adding gelation agents, such as salt, acid, and enzymes, to an emulsion to create a gel structure. This molding method usually requires preheating treatments. Unlike thermal induction, the gelation process does not proceed during the preheating stage but only after salt, acid, or enzyme addition. The unfolded protein remains soluble [26], while preheating enables the denatured unfolding of the protein in response to heat. Furthermore, it is not easy to deactivate the thermally unstable bioactive substances in the emulsion in the pre-heating process, causing loss of function [27]. During acid-induced gelation, the acidification caused by glucono-δ-lactone addition decreases the pH and neutralizes the surface charges of the protein aggregates, leading to gel structure formation via Van der Waals forces and hydrophobic interaction [18]. Salt induction usually comprises the addition of Ca^2+^ from CaCl_2_, which changes the pH and ionic strength to reduce the electrostatic repulsion between the proteins. This process promotes ionic cross-linking, allowing protein aggregates to form gel networks, increasing the ionic strength, and accelerating protein aggregation to generate larger aggregates [28]. Enzyme induction involves the addition of enzymes to facilitate the covalent cross-linking of proteins [29]. This process typically uses microbially derived transglutaminase (TG) to catalyze acyl transfer reactions, deamidation, and cross-linking between intra- or interchain protein glutamine and lysine peptide residues [30]. Previous studies have shown that the gel obtained using TG as a gelation agent exhibits a delicate, robust texture and higher mechanical strength than when using other methods [31]. Enzyme induction presents various advantages, such as mild and controllable gelation conditions and no by-products [32]. Furthermore, research has shown that ethanol is feasible in inducing the gelation of emulsion condensation [27], presenting a novel method for preparing a cold-set gel while providing insight into the development of new functional foods.

As rigid, polydisperse, hydrophilic macromolecules, polysaccharides can be used for thickening and gelling aqueous media to obtain polysaccharide-based emulsion gels. Non-starch polysaccharides can usually be gelatinized by cooling, heating, or adding calcium ions [21]. In experimental and practical applications, polysaccharide-based cold-gel agents, such as sodium alginate, xanthan gum, and carrageenan, have often been chosen to prepare polysaccharide-based emulsion gels or improve the properties of protein gels. These macromolecular species can facilitate polymer interaction to form continuous networks with emulsion gel functionality [16]. In practical situations, the specific mode of preparation depends on the matrix of the emulsion gel and its application in food.

## 3. The Application of Emulsion Gels as a Fat Replacement

Emulsion gels with soft, solid textures are more suitable for developing as fat substitutes than traditional emulsions without gel formation. They can better mimic the physical properties of animal fats like pork backfat, such as hardness and water retention capacity. Since these emulsions are more suitable for transporting and protecting oxidized lipids in food [33] and are more successful in preserving flavor substances and bioactive compounds, they can be used to improve the nutritional characteristics of meat products.

A variety of formulations, proteins, polysaccharides, and low-molecular-weight compounds can be used as emulsion gel matrices [18]. These polymer molecules are held together by weak intermolecular forces (e.g., hydrogen bonding, electrostatic forces, Van der Waals forces, and hydrophobic interactions), while covalent bonds (disulfide bonds) are also involved in the heat-induced gelation of globular protein gels [34]. Emulsion gel preparation usually involves the production of a protein-stabilized emulsion, which may also be supplemented with a hydrocolloid stabilizer or other ingredients (proteins, polysaccharides, and surfactants) after emulsion formation [35]. Proteins and polysaccharides are often used as gelation agents in emulsion gels used as fat substitutes, which are broadly classified as protein-, polysaccharide-, and protein–polysaccharide composite-based emulsion gels, depending on the matrix. The application of emulsion gels as fat substitutes in different meat products is summarized in Table 1.

### 3.1. Preparation of the Emulsion Gel

#### 3.1.1. The Gelation Mechanism and Influencing Factors

Proteins contain many functional groups that can be covalently cross-linked and are ideal for emulsion gel formation [36]. Combining different types of proteins may result in emulsion gels with different properties and expanding their applications in food products [13]. Emulsion-filled protein gels contain proteins that can both stabilize the emulsions as emulsifiers and form network structures as gelation agents [26]. The most important concerns during the preparation of protein-based emulsion gels are their rheological properties and network structures, the gelation properties of which are mainly determined by the protein characteristics and concentrations, the oil droplet content, and the heating temperature and time [26], as well as other factors, such as pH, ionic strength, and gelation temperature [25].

The protein-based emulsion gels used in meat products frequently contain soy proteins and sodium caseinate (SC) due to their high nutritional value and emulsification, thickening, and gelation properties [37,38]. Soybean protein provides surfactant molecules, reduces the interfacial tension between oil and water, and enhances the stability of emulsion gels [39]. Studies have shown that soybean protein isolate (SPI) can be employed to produce emulsion gels with excellent freezing-thawing stability and rheological properties using NaCl [28], increasing their potential as fat substitutes. However, compared with polysaccharide and protein–polysaccharide composite-based emulsion gels, it seems more difficult to produce emulsion gels with a certain hardness and gel strength using proteins. Marie-Christin Baune et al. [40] hypothesized that a higher internal phase (oil) content (above 50%) and protein concentration could increase the rigidity and viscoelasticity of emulsion gels. They attempted to identify a commercial separating protein from soy, pea, or potato suitable for preparing pH-neutral (6.5) and heat-resistant (72 °C) emulsion gels for use as solid animal fat substitutes. The experiments indicated that both the interfacial and protein–protein interactions of leguminous proteins were involved in structural reinforcement, while the hardness increased with the cysteine content, and the interactions displayed electrostatic, hydrophobic, and hydrophilic properties. Leguminous proteins seem to hold more promise for preparing stable, solid animal fat substitutes suitable for long-term storage than potato proteins.

#### 3.1.2. Protein-Based Emulsion Gels as Fat Substitutes in Meat Products

(1)Reducing the fat content and improving the fatty acid profile

Several studies have examined the addition of protein-based emulsion gels to meat products as fat substitutes. Pintado et al. [41] used olive oil and chia oil as raw materials to structurally prepare oleogels, while SPI and gelation agents (gelatin) were added to prepare emulsion gels. Oleogels are a type of organogels, which can be defined as a three-dimensional cooling network structure with thermally reversible properties formed by an organic liquid and organogelators. When the organic phase of an organogel is edible oil, it is called an oleogel. Unlike emulsion gels, which are formed by making the continuous phase gelation, oleogels are liquid oils that are transformed to a gel state by organogelators, such as beeswax, which consist mainly of oil and have a much higher oil content than emulsion gels [42,43]. In this study, both gels were used as fat substitutes, and their suitability for replacing functional fermented meat product (fuet) was evaluated. The results indicated that fat substitution improved the fatty acid composition of the products, with a 12-fold decrease in the unsaturated fatty acid n-6/n-3 ratio compared to the control samples with normal fat content and reduced fat content (more water), while the products exhibited an excellent oxidative and microbial status during a 30-d frozen storage period. Therefore, both the emulsion gel and oleogel could be used as meat product fat substitutes.

Various studies have explored the gelation mechanism of protein-based emulsion gels and new gel preparation methods to obtain fat substitutes with physical, chemical, and sensory properties similar to real animal fats. Dreher et al. [11] suggested that a certain amount of solid fat could increase the similarity of the key properties of fat mimetics derived from plant materials to animal adipose tissue. Melted solid-state hydrogenated canola oil was mixed with liquid canola oil and added to a solution containing an excess of hot emulsified SPI, which was covalently cross-linked using TG to produce a protein network. The internal solid fat was allowed to crystallize to form additional networks, ultimately resulting in an integral emulsion gel with melting and elastic properties. The results indicated that the emulsification of a lipid crystal network consisting of liquid plant oil and solid plant-derived fat with excess plant proteins produced different structures via TG-induced cross-linking. Furthermore, plant-derived lipids could be engineered to create fat crystal networks that mimic the mechanical properties of animal fats via sequential melting, emulsification, cooling, and cross-linking.

In addition, several studies [12,44,45,46] explored the effect of the protein content on plant-based emulsified cross-linked fat crystal networks simulating animal adipose tissue, confirming that the textural and rheological properties of emulsion gels can be changed to some extent by modifying the protein content in the initial emulsion before TG cross-linking.

(2)Structured liquid oils

Plant or marine oils rich in unsaturated fatty acids are primarily in a liquid state. The introduction of these liquid oils with completely different physicochemical properties from solid animal fats can negatively affect the quality of the product. Liquid oils can be modified or structured to present a healthy fatty acid composition while maintaining their solid characteristics and plasticity [19]. Emulsion gels can be used for liquid oil structuring, providing a strategy for introducing healthier oils into reformulated low-fat meat products. Since liquid oils are stabilized in a gel network composed of a protein matrix, the O/W emulsion gel containing non-meat proteins improves the fat binding capacity of the system, stabilizing the oil in the structures of the meat products. Several studies [47,48,49] introduced healthy oil combinations consisting of olive, linseed, and fish oil, into Frankfurters, using different protein-stabilized O/W emulsions. Furthermore, TG enzymes were incorporated to facilitate gel structure formation, yielding low-fat meat products with low SFA and high PUFA levels and suitable n-6/n-3 ratios that displayed consumer acceptability, indicating that this was a feasible method for stabilizing liquid oils.

(3)The introduction of functional ingredients and application of new technologies

Compared with emulsions, oleogels, and hydrogels [50], emulsion gels are a better choice for improving the nutritional properties of food by carrying hydrophobic compounds and functional ingredients and protect bioactive compounds that act as carriers for hydrophobic compounds and functional ingredients and protect bioactive compounds within them [51]. Compounds, such as polyphenols, display antioxidant activity and are often artificially added to improve the nutritional properties of foods. However, directly adding polyphenols to meat products degrades or inactivates them and may also cause a decline in the quality of the color and taste.

Freire et al. [52] formulated emulsions using different protein emulsifiers and a lipid phase rich in n-3 unsaturated fatty acids. They produced cold-set gels after adding a natural extract rich in concentrated tannins (CT), increasing the antioxidant activity and stability. Pintado et al. [53] used extra virgin olive oil and SPI to produce emulsion gels as a phenolic compound delivery system to provide the product with moderate amounts of polyphenols to take advantage of their health benefits. Therefore, the use of emulsion gels containing two different polyphenol extracts (grapeseed or grapeseed and olive) as fat substitutes in Frankfurters was evaluated. The results indicated that incorporating the emulsion gel reduced the SFA proportion in the product by half with a higher PUFA/SFA ratio. Moreover, the Frankfurters containing the emulsion gel with the solid polyphenol extract displayed high levels of hydroxytyrosol (Hxt) and gallic acid, flavanol monomers, and their derivatives, while the trophic advantage did not cause undesirable sensory or structural changes and provided excellent stability. Therefore, the emulsion gel is suitable as a release system for polyphenols without obviously influencing the sensory properties of the product.

Shahbazi et al. [54,55] used SPI and canola oil as raw materials to prepare emulsions. Several different biosurfactant agents (ethyl cellulose (EHEC), octenyl succinic anhydride (OSA) starch, acetylated starch, and dodecyl succinate (DS) inulin) were separately added to the emulsion to replace some or all of the oil to prepare the desired low-fat soy protein emulsion gel, which was applied for 3D printing. Consequently, the emulsion gel was suitable for ink development when 3D printing low-fat artificial meat, providing a method for developing 3D-printed plant-based meat products.

### 3.2. Polysaccharide-Based Emulsion Gels

#### 3.2.1. Gelation Mechanism and Influencing Factors

When preparing emulsion gels as fat substitutes, it is necessary to maintain the desired appearance and rheological characteristics and acceptable sensory properties of the meat products after gel incorporation. An increase in the fat substitute proportion is generally associated with a decrease in sensory quality, especially a deterioration in juiciness, while polysaccharide gels display a higher water-retention ability and are suitable for constructing gels with different properties. Lipophilic agents with beneficial health effects can be incorporated into food while maintaining the product characteristics [33].

During the preparation process of polysaccharide-based emulsion gel, the polysaccharides are dissolved at a high temperature, the emulsions are prepared at a moderate temperature, and gels are formed at a low temperature. The gelation mechanisms include the formation of double and cross-linked helical domains during cooling to form 3D structures [18]. The advantages of polysaccharides include their ability to control food texture and flavor release [34], while their diverse structures and gelation conditions show significant promise for tailoring gels with desirable structures [56]. Characteristics such as molecular weight, size, monosaccharide composition, charge density, molecular conformation, and extrinsic conditions, such as temperature, pH, and ionic strength, are important factors affecting the structures of polysaccharide gels. Polysaccharides such as agar, carrageenan, pectin and konjac, are also important gelling agents in food, and they can also be considered as gel matrices [43]. Polysaccharides can form gels at concentrations below 1% via various molecular interactions [57].

#### 3.2.2. Polysaccharide-Based Emulsion Gels as Fat Substitutes in Meat Products

Inulin and chia powder are often used as raw materials for polysaccharide-based emulsion gels due to their rich dietary fiber and functional properties, such as gelling, emulsification, and fat water-binding abilities [58,59]. In addition to its gelation, thickening, and stability properties [60], carrageenan can also act as a meat binder and a texture stabilizer [37]. Some studies indicated that adding dietary fiber to meat products can help improve the stability and rheological properties of emulsions [61]. Many polysaccharide biopolymers, such as konjac, inulin, and carrageenan, are commonly used to develop fat substitutes and have yielded excellent results thus far. Konjac glucomannan, extracted from the east Asian native plant, konjac, can be used for gel formation and fat substitution. Konjac gels can be ground to the desired particle size to simulate visible granular fat, which is suitable for mimicking the sensory properties and replacing animal fat [62].

(1)Reducing the fat content and improving the fatty acid profile

Using polysaccharide emulsion gels as fat replacers can improve the water retention of meat products and promote regular water release during fermentation, which can maintain the sensory properties of products, such as dried fermented sausages [63]. Alejandre et al. [64] prepared high-ω-3-content emulsion gels using κ-carrageenan and linseed oil. The phases were heated separately to 70 °C and emulsified via homogenization. Emulsion gel formation occurred via κ-carrageenan polymerization after gel cooling, allowing its application in dry-fermented sausages as a fat replacement. Compared with the control group, this method improved the fatty acid composition of dry-fermented sausages, reducing the ω-6/ω-3 ratio. The addition of this emulsion gel increased the α-linolenic and ω-3 unsaturated fatty acid content, while no significant differences were evident in the color, taste, and juiciness of the reformulated low-fat sausages compared to traditional dry-fermented sausages. Therefore, emulsion gels prepared using linseed oil and carrageenan can be used as an alternative to dry-fermented sausages, since they retain sensory qualities acceptable to consumers.

(2)Structured liquid oils

Even though not prepared as an emulsion gel, gels produced using only polysaccharides can also be used as fat replacers. Although Ruiz-Capillas et al. [62] effectively reduced the fat in dry-fermented sausages using konjac gel as a fat substitute, this technique decreased the quality of the products to some extent. However, since the properties of emulsion gels differ from standard polysaccharide gels, liquid plant oils can be introduced into meat systems. Adding healthier plant oils may improve the fatty acid profiles in meat products while reducing the fat content. However, healthy plant oil rich in unsaturated fatty acids is prone to accelerated oxidative deterioration and shorter shelf life of foods while reducing the plasticity of the final product, causing a textural decrease and loss of nutritional properties. Therefore, it is necessary to stabilize the emulsion by reinforcing the structure. Structured liquid oil converted into emulsion gels can produce rheological properties close to those of animal fat [39,64], showing promise as a fat replacement approach when designing and developing low-fat meat products. Alejandre et al. [42] compared an organogel and an emulsion gel with κ-carrageenan as a matrix to examine the effect of animal fat replacement in these two structured oil systems in meat batter. The results indicated that meat batters formulated using organogels exhibited higher matrix stability and were incorporated into the meat matrix more efficiently than meat batter formulated with emulsion gels. However, the emulsion gel showed sufficient performance as a structured oil method, compensating for the deficiencies of direct plant oil addition. Therefore, it could be used as a fat substitute in the meat batter. Many studies have mixed olive oil, linseed oil, and other healthy oils to create stable emulsion gels with polysaccharide matrices, which were added to Frankfurters and other meat products to replace the fat, achieving relatively favorable results [65,66,67,68].

(3)The development of fat cube substitutes

Polysaccharide-based emulsion gels can be converted into cubes to provide the appearance of visible fat lumps in meat products [69], such as sausages, consisting of a mixture of solid lumps of pork backfat and lean meat. However, studies involving simulated solid fat cubes and emulsion gels indicated that the emulsion gel must display a certain hardness and strength. Chen et al. [63] prepared fat cube substitutes using konjac glucomannan and κ -carrageenan to partially replace the pork backfat in dry Harbin sausages. The results showed that although using solid cube fat substitutes prepared via an emulsion gel was a feasible method for reducing the fat content in dry-fermented Harbin sausages, the property changes were related to the substitution level. No significant differences were evident between the physicochemical and sensory properties of the lower substitution level group and the control group, while high substitution levels produced changes in product characteristics. To ensure the sensory attributes of the dry Harbin sausages, the upper limit of the cube fat substitution level was 40%.

(4)The introduction of functional components

As with their protein-based counterpart, polysaccharide-based emulsion gels can also be used to protect active substances and improve the nutritional value of food products. Alejandre et al. [70] added catechin-rich natural extracts from blackthorn branches to a κ-carrageenan emulsion gel system containing microalgal oil to obtain functional ingredients and use it as a fat substitute in beef patties. The results indicated that adding the extract to the polysaccharide-based emulsion gel system did not affect the overall sensory properties and acceptability of the product. Moreover, the extract provided high antioxidant properties to the emulsion gel, decreased the fat content in the beef patties after fat substitution, doubled the antioxidant activity and DHA content, and increased the antioxidant stability by reducing peroxide content.

### 3.3. Protein–Polysaccharide Composite-Based Emulsion Gels

#### 3.3.1. The Gelation Mechanism and Influencing Factors

Combining proteins and polysaccharides represents a new approach for developing novel solid fat mimetics. Compared with the other two matrices, the composite matrix has been studied more extensively and presents a broader application scope. Emulsion gels containing polysaccharides and proteins display a higher similarity to real gel systems in food [43]. The simultaneous addition of protein and polysaccharide macromolecules may cause intermolecular correlations via electrostatic interaction, complex coacervation, and associative phase separation [71] to generate complexes that can quickly form network structures in food systems and produce diverse functions [72]. This method is commonly used to control the structure of food products for better texture and stability, while the diverse structures and gel conditions of polysaccharides make it possible to tailor gels with desirable structures [56].

Protein–polysaccharide gels are produced via gelation that does not require a denaturing process during protein gel preparation. The porosity and structures of the gels can be adjusted by changing conditions, such as protein or polysaccharide species, added quantity, ratio, pH, and salt concentration, while potentially altering the rheological properties and stabilizing the emulsions [57]. After a protein-based emulsion gel is generated, the hydrocolloids and other natural ingredients formed by the polysaccharides can also be added to tailor the gel structure and functionality. Furthermore, adding flaxseed gum changes the rheological properties of peanut protein isolate emulsions and gels, acts as a thickener to reduce gelation time, and improves gel strength [72]. Flaxseed gum or flax gum addition also enhances the apparent viscosity of SPI emulsion [73], playing a crucial role in improving the emulsion gelation properties, enhancing thermal stability, and increasing the structural strength of the gel network [74]. Therefore, adding polysaccharides to the aqueous phase of emulsions can act as a thickener, improving emulsion instability.

#### 3.3.2. Protein–Polysaccharide Composite-Based Emulsion Gels as Fat Substitutes in Meat Products

(1)Reducing the fat content and improving the fatty acid profile

Protein–polysaccharide composites display better functionality than individually acting proteins or polysaccharides. Santos et al. [75] used pork skin, inulin, α-cyclodextrin, and bamboo fiber as raw materials to replace the pork backfat in emulsified meat products. The pork skin, rich in collagen, exhibited high gelation and emulsification capacity. Furthermore, the interaction between the pork skin and dietary fiber significantly improved the hardness and stability of the emulsion gel, while the addition of bamboo fiber also enhanced the performance of the emulsion gel to a certain extent. The application scope of composite-based emulsion gels is broader than those with a single matrix. They are used in meat products, such as sausages and burger patties, as well as for developing seafood analogs. Modifying their concentrations can control the texture and network structures of the gels. Ran et al. [76] used konjac glucomannan to enhance cross-linking with soy protein to mimic the texture of fish balls. According to the results, konjac glucomannan addition significantly affected the textural and rheological properties of plant-based fish balls while increasing the hardness, chewiness, and gel strength at a higher concentration. The addition of konjac glucomannan contributes to the formation of a tighter gel network and denser cross-linking. A low konjac concentration increases the porosity and density of the structure, while an appropriate concentration enhances elasticity and gel strength. Therefore, the polysaccharide–protein composite-based emulsion gel displays excellent potential and application value for developing plant-based seafood analogs and novel low-fat meat products.

(2)The development of fat cube substitutes

Both the strength and hardness of the gels can be improved by the combined effect of polysaccharides and proteins. This change is beneficial to the formation of the 3D cube fat substitute, modifying the textural properties and gel network structures of emulsion gels by adjusting the number of added polysaccharides and proteins, facilitating customization to obtain the optimal emulsion gel structure. Huang et al. [46] used an aqueous SPI solution homogenized with coconut oil to prepare an emulsion and subsequently added konjac glucomannan to obtain a 3D cube emulsion gel with a certain hardness and strength via the TG cross-linking effect. This was used as a fat substitute to study the effect of adding different protein and konjac concentrations to the emulsion gel systems. The findings indicated that the emulsion gel could be used to simulate a solid cube fat. It was similar to pork fat in appearance, exhibited desirable functional qualities in terms of both mechanical and oral tribological properties, and was favored by consumers at an added protein content of 1% and konjac content of 4%. The 3D structures of fat substitutes can provide the product with the desired visible fat cube appearance, for which a protein–polysaccharide composite matrix offers a potential approach. Moreover, protein–polysaccharide complexes are more successful in improving the performance deficit caused by a single raw material matrix than proteins or polysaccharides alone.

(3)The application of cereal flour as a composite matrix emulsion gel

In addition to dietary fiber and protein, some cereal powder products also contain a high number of bioactive compounds, such as cocoa bean shells rich in catechins, theobromine, and caffeine, which can be used as matrices for developing emulsion gels [77]. Pintado et al. [78] used emulsion gels as delivery systems for healthier bioactive plant compounds to prepare gels from chia powder and olive oil as fat substitutes in low-fat Frankfurters. The results showed that adding chia powder emulsion gel reduced the fat level to 40%, consequently decreasing the energy intake by 30%, which could be labeled as “reduced fat content”. The reformulated Frankfurter contained a large number of minerals, such as magnesium, manganese, and calcium, with more lipid–protein interactions. Adding the emulsion gel retained the sensory characteristics of the Frankfurters within the accepted range, while the product displayed oxidative stability during storage. Subsequent studies used chia and oat-based emulsion gels to replace animal fats in low-fat fresh sausages (longanizas) [79,80]. This reduced the product fat and energy, improved the fatty acid ratio, decreased cooking loss, and increased the concentrations of certain minerals and amino acids. In addition, oat emulsion gels are used as fat substitutes to provide meat products with β-glucan and monounsaturated fatty acids (MUFA) [81]. Therefore, chia or oat emulsion gels as animal fat substitutes may increase the nutritional value of meat products, such as Frankfurters.

(4)Changes in the sensory properties of products caused by different substitution ratios

Due to the role of fat in meat products, changes in the sensory properties limit the application of fat substitutes. Therefore, meat products with an appropriate proportion of emulsion gel yielded mostly desirable results in sensory evaluation tests, while 100% animal fat substitution usually produced various negative effects. Serdaroğlu et al. [82] showed that completely replacing the beef fat in chicken patties with emulsion gels produced the lowest sensory evaluation scores, while samples with 25% and 50% fat substitution exhibited similar scores to the whole fat samples. The results indicated that adding 50% emulsion gel rendered the chicken patties similar to the original product. The beef fat played an important role in flavor and provided the product with its characteristic flavor, even when reduced by half. Studies have shown that replacing all the fat with emulsion gels negatively impacts meat products [16,78]. In most studies, the meat products in which 50% of the fat was replaced by emulsion gel displayed the best comprehensive quality, while a total substitution decreased the sensory properties of the samples while negatively affecting the textural and technological performance [83]. However, Berker Nacak et al. [84] produced composite emulsion gels using peanut and linseed oil to replace beef fat. These samples showed higher oil scores and overall acceptability during the sensory evaluation than the whole-fat control specimens. This result could be attributed to the emulsion gel providing the desired taste by covering the oil beads and their characteristic solid-like structures, consequently displaying a high simulation capacity. The different results may be related to the selection of raw materials, preparation technology, and the application of emulsion gels in different meat products. However, emulsion gels can potentially reduce the fat content of meat products and improve the distribution of fatty acids without reducing the sensory quality.

**Table 1 foods-11-01950-t001:** The application of emulsion gels with different matrices as fat substitutes in meat products.

The Matrix Type	Oil	Protein	Polysaccharide	Gel Method	The Application of Fat Substitutes	References
Protein-based	Olive oil	SPI, SC	-	TG induction	Pork backfat in Frankfurters	[85]
	Olive oil, linseed oil, fish oil	SPI, SC	-	TG induction	Technical applicability in different meat products	[49]
	Olive oil, linseed oil, fish oil	SPI, SC	-	TG induction	Pork backfat in Frankfurters	[48]
	Olive oil, linseed oil, fish oil	SPI, SC	-	TG induction	Pork backfat in Frankfurters	[47]
	Olive oil, chia oil	SPI	-	Gelatin was used as a gelling agent	Pork fat in functional fermented meat products (fuet)	[41]
	Canola oil	SPI	-	Adding hydrophobically modified biosurfactants (acetylated starch, OSA starch, EHEC, and DS inulin)	Low-fat meat product simulant constructed by 3D printing	[54,55]
	Olive oil	SPI	-	Gelling agent based on alginate	Fat in Frankfurters and research on the development of meat products rich in polyphenols	[53]
	Canola oil, sal fat, or mixtures thereof	SPI		TG induction	Fat in salami-type sausage	[86]
Polysaccharide-based	Olive oil, linseed oil, fish oil	-	Konjac glucomannan, i-carrageenan, pre-gelled corn starch	Ca(OH)_2_ induction	Pork backfat in low-fat pork liver pâté	[65]
	Olive oil, linseed oil, fish oil	-	Konjac flour, pre-gelled corn starch	Ca(OH)_2_ induction	Pork backfat in dry fermented sausage	[66]
	Olive oil, linseed oil, fish oil	-	Konjac flour, pre-gelled corn starch	Ca(OH)_2_ induction	Pork backfat in Frankfurters	[67]
	Olive oil	-	Konjac flour, i- carrageenan, pre-gelled corn starch	Ca(OH)_2_ induction	Beef fat in fresh sausage (Merguez)	[87]
	Olive oil, linseed oil, fish oil	-	Konjac flour, pre-gelled corn starch, ι- carrageenan	Ca(OH)_2_ induction	Pork fat in dry-fermented sausage	[68]
	Linseed oil	-	Carrageenan	Cooling after heating	Part of the fat in Bologna sausage	[88]
	Sunflower oil	-	Carrageenan	Cooling after heating	Pork backfat in burger patties	[33]
	Soybean oil	-	Aloe gel	Cooling after heating	Fat in low-fat meat emulsion	[89]
	Linseed oil	-	κ-carrageenan	Cooling after heating	Pork backfat in dry-fermented sausage	[64]
	Algae oil	-	κ-carrageenan	Cold induction	Fat in beef patties	[90]
	Linseed oil	-	inulin	pork gelatin was used as a gelling agent	Pork backfat in dry-fermented sausage	[91]
	Canola oil	-	κ-carrageenan	Cooling after heating	Fat in beef batter	[42]
	Microalgal oil	-	κ-carrageenan	Cold induction	Fat in beef patties	[70]
	Chia oil, linseed oil	-	κ-carrageenan	Heat induction	Pork backfat in low-fat burgers	[92]
	Pre-emulsified olive oil	-	Jerusalem artichoke powder	Heat induction	Pork back fat in Harbin dry sausages	[93]
	Corn germ oil	-	Konjac flour, κ-carrageenan, barley β glucan	Heat induction	Pork backfat in Harbin dry sausages (cube fat substitute)	[63]
	Canola oil, olive oil	-	κ-carrageenan	Carrageenan was used as a gelling agent.	Pork backfat in beef burgers	[94]
Protein–polysaccharide composite-based	Olive oil	Chia powder (protein 22%)	Chia powder (dietary fiber 30.2%)	Three different cold-gelling agents: MTG and caseinate; sodium alginate, CaSO4, and pyrophosphate; gelatin	Pork backfat in low-fat Frankfurters	[78]
	Olive oil	Gelatin	Inulin	Gelatin and inulin were used as gel agent	Beef fat in model system meat emulsion	[83]
	Olive oil	Chia powder	Chia powder	Gelling agent based on alginate	Pork backfat in low-fat Frankfurters	[35]
	Olive oil	Chia powder	Chia powder	Gelling agent based on alginate	Pork backfat in low-fat Frankfurters	[95]
	Olive oil	Gelatin	Inulin	Cooling after heating	Beef fat in chicken patties	[82]
	Linseed oil	Gelatin	Inulin	Cooling after heating	Beef fat in the model system meat emulsion	[96]
	Olive oil	Gelatin	Inulin	Cooling after heating	Beef fat in model turkey breast emulsion	[97]
	Olive oil	Chia powder, oat bran (protein 20%)	Chia powder, oat bran (dietary fiber 44%)	Gelling agent based on alginate	Fat in low-fat fresh sausage (longaniza)	[79]
	Soybean oil	SPI, SC	Carrageenan, pectin, inulin	Heat induction	Pork backfat in meat products develops functional emulsion gels	[37]
	Camellia oil	SC	κ-carrageenan	Cold induction	Pork fat in low-fat Harbin sausage	[98]
	Soybean oil	SPI, SC	Chia powder, inulin, carrageenan	Heat induction	Pork backfat in Bologna sausage	[39]
	Soybean oil	SPI	κ-carrageenan, ι-carrageenan, inulin,	Heat induction	Pork backfat in low-fat Frankfurter sausage	[51]
	Canola oil	Pork skin (protein 37.7% ± 0.9)	bamboo fiber, inulin, polydextrose, α-cyclodextrin	Cooling after heating	Pork backfat in emulsified meat products	[75]
	Olive oil	Chia mucilage,	Chia mucilage	Gelling agent based on alginate	Pork backfat in Bologna sausage	[99]
	Soybean oil	SPI	Inulin, carrageenan	Carrageenan was used as a gelling agent	Pork backfat in Bologna sausage	[100]
	Olive oil	Chia mucilage, whey protein, collagen	Chia mucilage	Six gelling agents (sodium alginate, collagen, whey protein, carboxymethylcellulose, TG, and carrageenan)	Pork backfat in emulsified meat products	[101]
	Black cumin oil, flaxseed oil	SC, gelatin	Inulin	Cooling after heating	Beef fat in functional fresh chicken sausages	[102]
	Peanut oil, linseed oil	Egg white albumin, gelatin	Inulin	Heating induction and cold induction were used, respectively	Beef fat in fermented beef sausage	[103]
	Olive oil	Chia mucilage, whey protein, collagen	Chia mucilage	Gelling agent based on alginate and collagen	Beef fat in fermented beef sausage	[104]
	Walnut oil	Cocoa bean shell flour (protein 17.13 g/100 g), gelatin	Cocoa bean shell flour (dietary fiber 61.18 g/100 g)	Gelatin was used as a gelling agent	Beef fat in fermented beef sausage	[77]
	Peanut oil, linseed oil	Egg white powder, gelatin	Chicory inulin	Gelatin was used as a gelling agent, cooling after heating	Beef fat in emulsified sausage	[84]
	Olive oil	Chia powder, oat bran	Chia powder, oat bran	Gelling agent based on alginate	Pork backfat in healthy fresh meat products (longanizas)	[80]
	Sunflower oil	SPI	Konjac glucomannan	Heat induction	Development of plant-based fish ball analogs	[76]
	Coconut oil	SPI	Konjac glucomannan	TG induction, heat induction	Simulated natural pork fat (3D structures)	[46]
	Soybean oil	SPI	Curdlan	Heat induction	Simulated pork backfat	[105]

SPI: soybean isolate protein; SC: sodium caseinate; TG: transglutaminase; the “-” means it does not contain this ingredient.

## 4. The Application of Double Emulsions for Fat Replacement

### 4.1. The Mechanism and Influencing Factors of Double Emulsions as Fat Substitutes

The development of emulsion gels as fat substitutes has attracted increasing research attention due to higher consumer interest in low-fat, functional meat products. The unique multi-compartment structures of double emulsions are more suitable than standard single emulsions and emulsion gels containing single emulsions in protecting sensitive bioactive compounds, providing a new technological strategy for developing functional low-fat meat products.

Most studies involving emulsion gels as a fat substitute use O/W emulsions as a base, while the utilization of inverse water-in-oil (W/O) emulsions is relatively low. Although Burcu et al. [106] considered the use of W/O emulsions to develop low-fat food formulations, encapsulate and control the release of bioactive compounds, and deliver healthy products to the target site, their study demonstrated that inverse emulsions could successfully reduce the total fat and SFAs in model meat batters. However, since the external phase of the inverse emulsion denotes an oil phase (O) rich in unsaturated fatty acids, which is prone to rapid oxidation during long-term storage and transportation, it must be combined with antioxidant substances to ensure product quality. Therefore, current research and applications using W/O emulsions in meat systems are limited.

The current use of double emulsions as fat substitutes in meat products is a promising direction. A double emulsion is a complex multidispersed, multiphase system, also known as a multiple emulsion, in which the O/W and W/O states co-exist. Furthermore, the dispersed emulsion is also an emulsion, known as an “emulsion in an emulsion” [107]. The water-in-oil-in-water (W/O/W) double emulsion contains two aqueous phases, namely the inner aqueous phase (W_1_) and the outer aqueous phase (W_2_). W_1_ is dispersed in the O to form W/O emulsions, which are distributed in W_2_ [107,108]. Therefore, there are two interfacial layers in the double emulsion: W_1_-O with a W_1_ and O, and O-W_2_ with an O and W_2_ [109]. Since the W_2_ includes O, the double emulsion compensates for the lack of oxidation in the W/O emulsion to a certain extent. However, some applications may also use oil-in-water-in-oil (O/W/O) emulsions [110].

### 4.2. The Application of Double Emulsions in the Food Industry

Double emulsions serve two primary purposes when utilized in food matrices. First, due to the unique partitioned structure of double emulsions, they show promise for separating, encapsulating, protecting, and promoting the sustained release and transportation of bioactive substances. Therefore, they can be used to encapsulate unstable ingredients, such as flavor, volatile, and bioactive substances, to disguise undesirable flavors in food and protect functional components. However, emulsion stability and controlling the release of these substances must be considered. The second aim is to produce double emulsions as fat substitutes for developing low-calorie, low-fat dairy and meat products [109]. Francisco [111] described the potential application of double emulsions for developing healthy and functional foods, proposing that these emulsions could qualitatively and quantitatively improve the fat quality of food by reducing the fat content and improving fatty acid distribution. The lipids in traditional O/W foods are partially replaced by water particle dispersion. Therefore, a double emulsion with the same total dispersed phase volume fraction and droplet size distribution as conventional O/W emulsions but a reduced fat content can be prepared to replace the fat in the product while still maintaining a similar texture and taste [107]. The application of double emulsions and gelled double emulsions as fat replacers in meat products is summarized in Table 2.

#### 4.2.1. The Application of Double Emulsions as Fat Substitutes in Meat Products

(1)The application of double emulsions to reduce fat and improve fatty acid composition

Although double emulsions can be used to replace the fat in meat products, their development is still in the early research stage. During the reformulation of meat products, the use of the same fat as the lipid phase to prepare a double emulsion is a potential method for developing low-fat meat products while ensuring that the physicochemical and sensory properties are similar to full-fat products. Freire et al. [112] used pig backfat as the lipid phase to prepare a W/O/W double emulsion for replacing the fat in Frankfurters. The fat content was significantly reduced (>60%), producing low-fat Frankfurters.

Using healthier plant oils as the lipid phase further improves the fatty acid composition in meat products while reducing their fat content. This approach is advantageous for developing low-fat meat products from a nutritional point of view. Cofrades et al. [113] prepared a double emulsion using olive oil as the lipid phase, which reduced the system fat content when used to replace pig backfat. This double emulsion showed excellent stability over a longer period, confirming the feasibility of using double emulsions as a technical approach to develop healthier meat products. In a subsequent study, Freire et al. [112] used a W/O/W emulsion prepared with perilla oil as the lipid phase to replace the pig backfat in Frankfurters. The results indicated a significant improvement in the fatty acid composition and reduction in the fat content, while adding perilla oil reduced the SFA and MUFA levels in the Frankfurters. The n-6/n-3 PUFA ratio decreased from 17 in the full-fat samples to 0.3 in the samples containing perilla oil as a fat substitute. Eisinaite et al. [114] encapsulated betaine as a colorant in W_1_ of the W/O/W emulsion and used it as a fat substitute in the meat system. It is possible to use oil to replace animal fat while avoiding the color change and quality reduction in meat products caused by the addition of double emulsion. This study demonstrates that these intrinsically stable emulsions can be incorporated into meat products without negatively affecting their textural properties, while encapsulating the colorants in the W_1_ of the double emulsion is a promising way to improve product color. Serdaroğlu et al. [109] used a W/O/W double emulsion prepared using olive oil as the lipid phase to replace the fat in a beef model system. The results indicated that the addition of a double emulsion significantly changed the textural parameters, evidenced by a decrease in hardness, gum viscosity, and chewiness. However, the total fat content in the system decreased, the fatty acid profile changed, and the total protein content increased, while the process and oxidation stability were enhanced. This provided new prospects for using double emulsions to produce low-fat beef products and healthier meat products. Therefore, double emulsions significantly reduce the fat content and improve the fatty acid compositions in meat systems.

(2)The examination of the oxidation stability

Oils high in unsaturated fatty acids can be used as the O in W/O/W double emulsions to develop healthy low-fat meat products. Unsaturated fats are prone to oxidation, resulting in nutrient loss and a decrease in the sensory characteristics of the product, which can negatively impact health, requiring antioxidative protection. Previous studies obtained different results regarding specific oxidation stability. Cofrades et al. [115] prepared a stable double emulsion with a homogeneous structure using chia oil while incorporating Hxt displaying antioxidant activity as a functional ingredient into W_1_ to evaluate the oxidative stability of this emulsion as a pork backfat substitute in meat systems. The results showed that chia oil rich in unsaturated fatty acids promoted lipid oxidation in cooked meat batter. The chia oil added to the double emulsion was more prone to oxidation than that added in a liquid form, while the thiobarbituric acid (TBA) values (reactants that reflect the degree of lipid oxidation) of the double emulsion samples exceeded those of the liquid chia oil specimens. This may be attributed to the more significant interaction between the meat components and oxidizable matrix, forming an interface surrounding the chia oil in the meat batter samples containing added liquid oil. In another related study [112], low-fat Frankfurters were produced using a double emulsion with perilla oil as the lipid phase. The results indicated that perilla oil, whether added directly or as a single or double emulsion, did not affect lipid oxidation when incorporated. However, Serdaroğlu [109] and Burcu Öztürk [116] conducted refrigerated stability tests on beef emulsion systems supplemented with double emulsions as fat substitutes. After 60 d of refrigerated storage, the TBA values of the double emulsion samples at different proportions were all lower than the control group prepared with 10% beef fat without a double emulsion. They proposed that although the presence of plant oils rich in unsaturated fatty acids was beneficial to lipid oxidation, the W_2_ acted as a continuous phase, prompting the formation of barriers covering the inner oil droplets that protected against lipid oxidation during storage. This variation in the experimental results may be ascribed to differences in the lipid sources, formulations, techniques, type of meat system, incorporation pattern characteristics of each oil, and its role in the meat protein matrix. However, various challenges remain due to minimal research involving double emulsions as fat substitutes.

(3)Adding functional components via double emulsions

Adding natural antioxidative compounds, such as polyphenols, to the primary W_1_/O emulsion of the double emulsion can prevent lipid oxidation during the O [117]. However, Cofrades et al. [115] showed that chia oil added in the form of double emulsions was more rapidly oxidized, while the oxidation stability of W_1_ was significantly improved after Hxt addition. The Hxt had a particularly significant effect on the oxidation stability of the system. Therefore, adding natural antioxidants to double emulsions is an effective way to address the rapid oxidative deterioration of meat systems. Studies have shown that lipophilic antioxidants can delay lipid oxidation in double emulsions more effectively than hydrophilic antioxidants [118].

While providing protection to oils, double emulsions also present a novel way to incorporate functional ingredients in meat products. Yogesh et al. [119] used Murraya koenigii berry extract, a natural antioxidant rich in phenolic compounds, as a W_1_ to prepare a double emulsion that was added to reduced-fat meat batter. The double emulsion stabilized the meat matrix and improved the oxidative stability of the reduced-fat meat batter after replacing the animal fat and did not negatively affect the other quality attributes of the meat batter. Encapsulating the pigment in W_1_ is feasible to improve the product color, since double emulsions can modify color characteristics. Eisinaite et al. [114] prepared a double emulsion using native beetroot juice as W_1_, which was added to a meat product system. The results indicated that the double emulsion reduced the heat load in the meat, facilitating maximum color retention even after heat treatment.

#### 4.2.2. Examination of the Gelled Double Emulsion as a Fat Substitute

Although double emulsions show significant potential for improving the fat content of meat products, providing bioactive substances, and disguising odors, their poor stability limits their application. During storage, double emulsions may exhibit creaming, phase inversion, phase separation, flocculation, and coalescence [120]. Gelation is a feasible method for improving stability [121,122]. The gelation of the inner aqueous droplets in the double emulsions via the incorporation of polymers or proteins into the inner droplets increases the stability against heat, shear, and the presence of salts. Yan Zhang et al. [123] prepared a gel-in-oil-in-water (G/O/W) emulsion and used it to develop low-fat emulsified sausages. This emulsion significantly reduced the fat content (about 40%) while maintaining similar textural characteristics to high pork oil emulsified sausages. Therefore, the G/O/W double emulsion was considered a suitable fat substitute. In addition, the role of lipid phase gelation in improving the stability of the double emulsion was investigated [124]. Biopolymers, such as polysaccharides, can stabilize the emulsion interface by adding them to the continuous phase, allowing their W_2_ to form a gel network that prevents creaming and coalescence, consequently improving the stability of double emulsions during storage (Figure 2) [125]. Gelled double emulsions are applied to encapsulate phenolic substances, such as green tea extract and Hxt [117,120], and can be used as fat replacers in meat products. However, minimal studies are available on this topic. Cofrades et al. [117] added perilla oil, which is rich in n-3 unsaturated fatty acids, as an O, and Hxt, a hydrophilic phenolic compound, as a natural antioxidant, to the W_1_. Gelatin and TG were added to the W_2_ of the double emulsion as gelation agents, which was used as a fat substitute in pork patties. The fatty acid composition, sensory properties, oxidative stability, and other properties of pork patties were determined. Although completely replacing the fat by gelled double emulsion decreased the acceptability of the product, the sensory quality of the patties with partial fat substitution was similar to that of the product with normal fat content. The fat levels in the substituted samples were significantly lower than those in the control group, while the SFA and MUFA (oleic acid) content decreased and PUFA significantly increased at a higher gelled double emulsion replacement ratio. Overall, the gelled double emulsion is a feasible strategy for developing healthier low-fat meat products.

However, compared with dairy products, not many studies exist involving fat substitution with double emulsions in meat products. Therefore, many challenges remain regarding the utilization of double emulsions in meat systems. In current studies, the double emulsion is mostly directly added to meat batter as a fat substitute but is rarely gelatinized. This may be due to the process complexity and post-processing instability of the gelled double emulsion, slowing down the research progress and application of this strategy. The emulsification process, W/O ratio, and types of surfactants can all affect the stability of the gelled double emulsion. Therefore, further research is required regarding the development and production of double emulsions and gelled double emulsions for low-fat meat products.

**Table 2 foods-11-01950-t002:** The application of double emulsions and gelled double emulsions as fat replacers in meat products.

W_1_	O	W_2_	Lipophilic Surfactant	Hydrophilic Emulsifiers	Research Content	References
NaCl	Olive oil	SC, whey protein concentrate, NaCl	Polyglycerol polyricinoleate (PGPR)	Whey protein concentrate, SC	Evaluates the effect of using double emulsion as a strategy to improve the fat content in meat model systems.	[113]
NaCl	Olive oil or pork lard	NaCl, SC	PGPR	SC	Possibility of food-grade double emulsion as a low-fat food ingredient in the meat industry.	[126]
NaCl, SC, Hxt	Chia oil	NaCl, SC	PGPR	SC	Antioxidant effect of double emulsion containing Hxt in the W_1_ as a fat substitute in a cooked meat system.	[115]
NaCl	Olive oil	NaCl, SC	PGPR	SC	Double emulsion replaces beef fat in a model beef emulsion system.	[109]
NaCl	Perilla oil, rendered pork backfat	NaCl, SC	PGPR	SC	Double emulsion replaces pork backfat in Frankfurters.	[112]
Native beetroot juice	Sunflower oil	Whey protein isolate	PGPR	Whey protein isolate	Double emulsion replaces pork backfat in the meat system and its effect on color improvement.	[114]
NaCl	Olive oil	Egg white powder, NaCl	PGPR	Egg white powder	Double emulsion replacement of beef fat in meat emulsion in the model system.	[116]
Hxt	Perilla oil	SC, gelatin, MTG	PGPR	SC	Gelled double emulsion replaces pork backfat in pork patties and its applicability as an n-3 PUFA and hydroxytyramol delivery system.	[117]
Olive leaf extract	Olive oil, linseed oil, fish oil	NaCl, SC	PGPR	SC	W_1_ contains double emulsion olive leaf extract to replace pork backfat in the meat system.	[127]
NaCl, bromelain	Canola oil	Tween^®^ 80	PGPR	Tween^®^ 80	Effect of bromelain liberated or embedded in the double emulsion on the eating palatability of pork loin.	[128]
NaCl	Olive oil	SC, whey protein concentrate, NaCl	PGPR	Whey protein concentrate, SC	Evaluates the effect of using double emulsion as a strategy to improve the fat content in meat model systems.	[119]
Gellan gum, CaCl_2_	Refined pork oil	SC	PGPR	SC	Development of low-fat emulsified sausages using G/O/W double emulsion.	[123]

## 5. Summary and Prospects

The results indicate that the emulsion gel can effectively reduce the fat content in meat products, improve the composition and proportion of fatty acids, and allow the products to be labeled as “reduced fat content”.

(1)Emulsion gels are suitable as fat substitutes

The choice of raw material is crucial for the structures of emulsion gels used as fat substitutes. The oil selected for the lipid phase, the protein and polysaccharide matrix interactions, and different gelation conditions make it possible to design a more ideal emulsion gel structure, affecting the structural and sensory properties of the product after incorporation into meat products. The use of fat substitutes in meat products can render them more compliant with the nutritional health needs of consumers by improving the fat content and fatty acid composition. When preparing emulsion gels as fat substitutes, liquid oil rich in unsaturated fatty acids can be selected as O. Plant oil with high unsaturated content is prone to oxidative deterioration. The characteristics of emulsion gels allow them to embed bioactive antioxidant substances in structured oil and enhance the stability of highly unsaturated oil. Furthermore, structured liquid oil converted into an emulsion gel can exhibit rheological properties similar to animal fat. The stabilization and structuring of liquid oils is a promising processing method achieved via emulsion gel preparation. Therefore, an emulsion gel provides a way to incorporate healthier oils into meat products.

(2)The development of solid cube fat mimetics

Adding specific emulsion gel proportions to meat products does not reduce the processing and sensory properties of the products or remains within the acceptable range for consumers. Completely replacing the fat using substitutes leads to a decline in the sensory properties. Although the emulsion gels prepared by some studies display gel structures, they exhibit high viscosity and a soft texture, making it difficult to obtain 3D cubes, while other research has shown that emulsion gels can simulate 3D fat cubes. In some meat products, such as sausages, well-separated cube fat particles are typically required to provide the desired mouth feel and visible fat cube appearance. However, the preparation of solid cube fat requires emulsion gels with a certain hardness and gel structure strength. This can be explored and developed in the future by changing the type and concentration of matrix raw materials and gel method.

(3)The development of double emulsions and gelled double emulsions as fat replacers

Double emulsions display unique separation structures that can be used to develop alternative fat while being capable of effective encapsulation to protect hydrophilic bioactive compounds, pigments, flavor substances, and enzymes. Although they are currently applied for researching fat substitutes in meat products, most are directly added to the model meat batter in the form of emulsions. The research on the application of gelled double emulsion is limited, while there is also a lack of studies on cube fat mimics, which may be related to poor stability. Double emulsions can be used to encapsulate the unpleasant flavors in fat substitutes or protect the active substances added to the W_1_ while avoiding the negative impact on sensory properties when added directly. Further research is required regarding the stability of double emulsions, especially the development of gelled double emulsions, which may lead to better sensory versus cooking properties of the fat substitutes in meat products.

## Figures and Tables

**Figure 1 foods-11-01950-f001:**
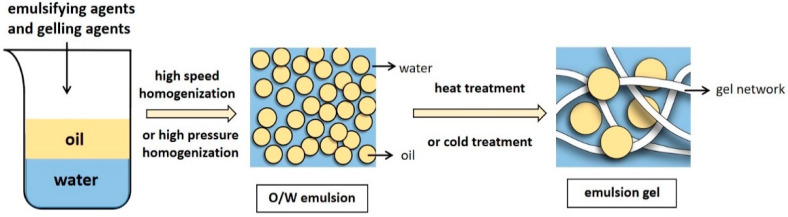
The preparation process of O/W emulsion gels.

**Figure 2 foods-11-01950-f002:**
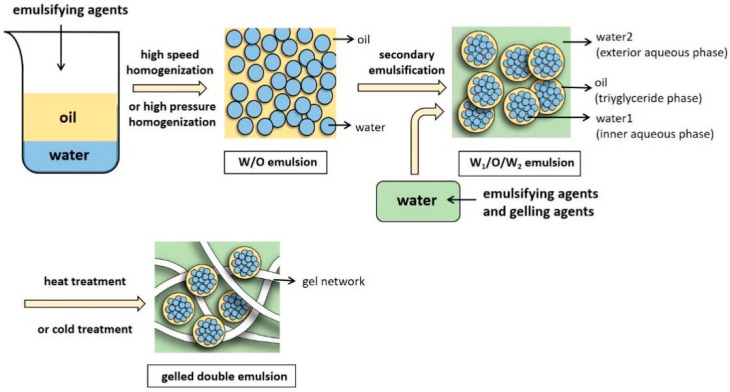
The preparation process of the W/O/W emulsion gel.

## Data Availability

No data were provided in the study.

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
