# Peer review of "Application of Emulsion Gels as Fat Substitutes in Meat Products"

_foods, 2022, doi:10.3390/foods11131950_

Round 1
Reviewer 1 Report
This review has comprehensively evaluated the application of emulsion gels as fat substitutes in meat products. The paper presents useful scientific information on an attractive area. I recommend the below-mentioned comments:
Line 27: reduced-fat
Line 79: Please define the oleogels, and organogels and their differences with emulsion gels in this section. (Both terms are used in the next sections)
Line 195: and expanding
Line 230: What was the control sample? (was the control sample full-fat meat product with animal fat or vegetable oil without gelation agent?)
Line 248: you have written “several studies” but only one reference is written. Please add more references
Lines 318-320: paraphrase
Line 327: delete “Therefore”, the previous sentence (reference 58) was about the incorporation of amaranth flour and is not related to these hydrocolloids.
Lines 322-333: These sentences are not about the gelation mechanism of polysaccharide-based emulsion gels
Line 339: high ω-3 content κ-carrageenan?! I think it is “high ω-3 content linseed oil”.
Reviewer 2 Report
This paper provides a detailed review on the application of emulsion gels in meat processing. This is an emerging area of study that many researchers around the world have devoted time and effort into researching. Overall, the review is well organized and written in an effective manner. The content covered in this review is comprehensive and is very complete.
I do think that some areas warrant improvement in terms of grammar and language. I would encourage the authors to read through the paper to ensure all details are correct.
Line-by-line suggestions:
Line 13: revise to “…such as lower levels of juiciness and hardness.”
Line 17: remove “-based”
Line 82: revise to “Emulsion gels, soft solid materials, are…”
Line 117: revise to “…step involves inducing the stable emulsion to an emulsion gel”
Line 136: revise to “…preheating treatments.”
Line 139: revise to “Furthermore, it is not easy to deactivate the…”
Line 157: revise to “…for preparing a cold-set gel…”
Line 162-164: awkward sentence, please rephrase this sentence. (sentence begins with “Therefore, in experimental and practical applications…”)
Line 258: revise to “Emulsion gels can be used…”
Line 270-272: awkward sentence, please rephrase this sentence. (sentence begins with “Compared with emulsions…”)
Line 460: revise to “…used to simulate a solid cube fat.”
Line 492: revise to “…chicken patties with emulsion gels.”
Table 1: format the first column so the words can be more easily read.
Line 522: revise to “Most studies involving emulsion gels as…”
Line 537: be consistent with the use of subscripts for W1 and W2 (this is also apparent on line 583, 606, 623, 632, 634, 648, 665, 671, 672, 737, and the headings of Table 2)
Line 583-586: awkward sentence, please rephrase this sentence (sentence begins with Eisinaite et al.)
Line 686: remove “In this study” this is a review paper and this phrasing is confusing.
